# In Vivo Treatment with Insulin-like Growth Factor 1 Reduces CCR5 Expression on Vaccine-Induced Activated CD4^+^ T-Cells

**DOI:** 10.3390/vaccines11111662

**Published:** 2023-10-30

**Authors:** Massimiliano Bissa, Veronica Galli, Luca Schifanella, Monica Vaccari, Mohammad Arif Rahman, Giacomo Gorini, Nicolò Binello, Sarkis Sarkis, Anna Gutowska, Isabela Silva de Castro, Melvin N. Doster, Ramona Moles, Guido Ferrari, Xiaoying Shen, Georgia D. Tomaras, David C. Montefiori, Kombo F. N’guessan, Dominic Paquin-Proulx, Pamela A. Kozlowski, David J. Venzon, Hyoyoung Choo-Wosoba, Matthew W. Breed, Joshua Kramer, Genoveffa Franchini

**Affiliations:** 1Animal Models and Retroviral Vaccines Section, National Cancer Institute, Bethesda, MD 20892, USA; 2Tulane National Primate Center & School of Medicine, Tulane University, Covington, LA 70118, USA; 3Division of Surgical Sciences, Department of Surgery, Duke University School of Medicine, Durham, NC 27710, USA; 4US Military HIV Research Program, Walter Reed Army Institute of Research, Silver Spring, MD 20910, USA; 5Henry M. Jackson Foundation for the Advancement of Military Medicine, Inc., Bethesda, MD 20817, USA; 6Department of Microbiology, Immunology and Parasitology, Louisiana State University Health Sciences Center, New Orleans, LA 70112, USA; 7Biostatistics and Data Management Section, Center for Cancer Research, National Cancer Institute, Bethesda, MD 20892, USA; 8Laboratory Animal Sciences Program, Leidos Biomedical Research Inc., Frederick National Laboratory, Frederick, MD 21701, USA

**Keywords:** IGF-1, HIV, SIV, insulin-like growth factor, CCR5, CD4, T-cells

## Abstract

At the heart of the DNA/ALVAC/gp120/alum vaccine’s efficacy in the absence of neutralizing antibodies is a delicate balance of pro- and anti-inflammatory immune responses that effectively decreases the risk of SIV_mac251_ acquisition in macaques. Vaccine efficacy is linked to antibodies recognizing the V2 helical conformation, DC-10 tolerogenic dendritic cells eliciting the clearance of apoptotic cells via efferocytosis, and CCR5 downregulation on vaccine-induced gut homing CD4^+^ cells. RAS activation is also linked to vaccine efficacy, which prompted the testing of IGF-1, a potent inducer of RAS activation with vaccination. We found that IGF-1 changed the hierarchy of V1/V2 epitope recognition and decreased both ADCC specific for helical V2 and efferocytosis. Remarkably, IGF-1 also reduced the expression of CCR5 on vaccine-induced CD4^+^ gut-homing T-cells, compensating for its negative effect on ADCC and efferocytosis and resulting in equivalent vaccine efficacy (71% with IGF-1 and 69% without).

## 1. Introduction

The severity of the HIV pandemic continues to grow on a massive scale as new infections outpace treatment options for affected populations. In 2021, approximately 1.5 million new HIV infections were recorded globally, yet only 75% of people living with the virus had access to antiretroviral therapy in that same year [1]. It is fully apparent that the development of an effective anti-HIV vaccine will be essential to control the pandemic. To date, the RV144 HIV Phase III vaccine trial concluded in Thailand in 2006 is the only trial to have provided limited but significant protection against HIV infection. Given the failure of several other candidate vaccines [2,3,4], improving on the RV144 vaccine modality is the clearest opportunity to potentially lead to a fully effective vaccine. An SIV-based vaccine mirroring RV144 in the macaque animal model demonstrated comparable vaccine efficacy in animals and reproduced the antibody response targeting V2 as a correlate of risk of SIV/HIV acquisition, demonstrating the model’s relevance to humans [5]. Interestingly, transcriptome and immune correlates analyses of reduced risk of SIV acquisition in macaques further identified a 12-gene signature that, at baseline, was predictive of protection. Ten genes out of the twelve identified belonged to the RAS pathway, and their expression level correlated with both anti-V2 antibody responses and with NK-p44^+^ IL-17^+^ cells, a mucosal subset of innate lymphoid cells also associated with decreased risk of virus acquisition [5]. Additional work in macaques with an improved RV144-like vaccine platform confirmed these two correlates and uncovered a key role of the monocyte CCL2/CCR2 axis and efferocytosis in vaccine protection [6].

The RAS superfamily comprises six families of small GTPases with over 150 members of small G proteins [7,8]. RAS operates mainly through two main cellular pathways: the mitogen-activated protein kinase (MAPK) pathway and the phosphoinositide-3 kinase (PI3K) pathway. RAS pathway activation is crucial for controlling cell growth and survival [9,10], contributing to the induction of both innate and adaptive immunity [7]. In monocytes, the ligation of CCL2 to its receptor CCR2 leads to RAS activation and subsequent integrin activation and chemotaxis [11,12]; in macrophages, the downregulation of MAPK and NF-κB pathways regulates M1/M2 polarization [13]; and in B-cells, RAS pathway activation occurs following B-cell receptor ligation [14]. KRAS also plays a crucial role in the development and maturation of B-cells through controlling the Raf-1/MEK/ERK pathway [15]. RAS isoforms in T-cells are activated after T-cell antigen receptor ligation, and the type of RAS isoforms can lead to development of different classes of T-helper cells [16].

Insulin-like growth factor 1 (IGF-1) binds to the IGF-1 receptor (IGF-1R) on the cell surface, causing conformational changes in the receptor and activating receptor tyrosine kinase activity [17]. This leads to the transmission of the signal through engaging numerous intracellular substrates and activating two major signaling pathways, the PI3K and the RAS/MAPK pathways [18]. The activation of these two pathways is required for the induction of different IGF-1 biological effects, such as cell proliferation, differentiation, and survival [17,18,19].

Due to the role of the expression of genes involved in RAS pathway activation and their association with correlates of reduced risk of SIV_mac251_ acquisition [5], we hypothesized that the induction of RAS pathway activation before and during immunizations could increase the efficacy of the anti-SIV DNA/ALVAC/gp120/alum vaccine strategy. To test this hypothesis, we searched the literature for possible RAS activators and selected IGF-1 as a safe and effective candidate to induce RAS activation, supported by its availability as recombinant protein as an FDA-approved drug for the treatment of growth deficit in children (Increlex^®^, Mecasermin, IPSEN).

IGF-1 affects cells of both the adaptive and innate immune system. B and T lymphocytes express IGF-1R [20], and their development and function are affected by IGF-1 [21,22,23]. Studies in mice revealed that IGF-1 treatment induced higher antibody levels [24], and in the absence of macrophage-derived IGF-1, the antibody titers elicited by subunit influenza vaccine were lower than in the wild-type condition [25]. Studies in human cells demonstrated that IGF-1 induced IgE and IgG4 [26]. IGF-1 stimulates both mouse and human regulatory T-cells, and following administration in mice, it suppresses autoimmune diseases such as type 1 diabetes and multiple sclerosis [27]. Monocytes treated with IGF-1 produced IL-10 and were able to suppress inflammation in mouse intestine [28].

Here, we tested whether the prime-boost anti-SIV vaccination administered with IGF-1 as an adjuvant affected vaccine immunogenicity and efficacy. We administered plasmid DNA encoding the macaque IGF-1 protein (DNA-IGF-1) and the human recombinant IGF-1 protein (Increlex^®^, Mecasermin, IPSEN) to macaques and observed a resultant increase in IGF-1 levels in plasma and increased activated RAS in plasma extracellular vesicles. Vaccinating macaques with the DNA/ALVAC/gp120/alum vaccine with or without DNA-IGF-1 plus Increlex^®^ increased some but not all immune responses correlated with decreased risk of infection [5,6,29,30]. The adjuvant did not significantly improve vaccine efficacy.

## 2. Materials and Methods

### 2.1. Animals, Vaccination, and SIV_mac251_ Challenge

This study enrolled Indian rhesus macaques (*Macaca mulatta*). Macaques were sourced by the National Institute of Child Health and Human Development (NICHD, Rockville, MD, USA), Primate Products Inc. (Immokalee, FL, USA), and Alpha Genesis Inc. (Yemasee, SC, USA). All animals were housed at the National Institutes of Health (Bethesda, MD, USA) and were handled in an AAALAC-accredited facility (OLAW, Animal Welfare Assurance A4149-01). Procedures and care of animals were conducted under an animal study protocol approved by the NCI Animal Care and Use Committee (ACUC). Macaques were monitored and cared for as previously described [6]. The animals were socially housed during the vaccination period and individually housed during the viral exposure period. A laboratory animal veterinarian supervised all clinical procedures, including the collection of biopsies, the administration of analgesics and anesthetics, and euthanasia.

#### 2.1.1. DNA-IGF-1 Administration Study

Four male (M) and five female (F) macaques were assigned to three groups of three animals each based on their sex, age, and weight. The three groups of animals received a single intramuscular injection of 1 mL of DNA containing 1000 (1 M and 2 F), 500 (2 M and 1 F), or 250 µg (1 M and 2 F) of DNA-IGF-1 in PBS. Following sedation, blood EDTA was collected at baseline and 1, 3, 6, 8, 10, 13, and 15 days following DNA administration.

#### 2.1.2. Increlex^®^ Administration Study

Four male and five female macaques were randomized into two groups accounting for sex, age, and weight. Five macaques (2 M and 3 F) received five doses of 0.08 mg/kg of Increlex^®^ diluted in PBS, whereas four macaques (2 M and 2 F) received five doses of 1 mL/each PBS. Increlex^®^ and PBS were administered subcutaneously at 2 h intervals without sedation. Following sedation, blood EDTA was collected prior to the first administration, and then 1 h following the third and fifth administrations.

#### 2.1.3. Vaccination Study

Fifty-two juvenile female macaques were divided into three groups accounting for their age, weight, and major histocompatibility status. The vaccinated group included one MamuA01^+^ and one MamuB17^+^ macaque, whereas the vaccinated+IGF-1 group included two MamuA01^+^ macaques.

Twenty-five macaques were intramuscularly vaccinated twice with DNA-SIV (weeks 0 and 4) as previously described [6,29]. Each DNA-SIV immunization contained 1 mg of 206S SIV p57gag_mac239_, 1 mg of 209S MCP3-p39gag_mac239_, and 2 mg of 221S SIV_macM766_ gp160. All twenty-five macaques were boosted twice (weeks 8 and 12) with the intramuscular administration of 10^8^ Plaque Forming Units of recombinant ALVAC (vCP2432, Sanofi Pasteur, Bridgewater, NJ, USA), which expresses SIV_mac251_ *gag-pro* and *gp120TM*. During the last immunization (week 12), macaques were administered 200 μg of SIV_mac251-M766_ and 200 μg of SIV_smE660-CG7V_ gp120-gD proteins. Protein boost was formulated in alum Alhydrogel (Invivogen, San Diego, CA, USA) as previously described [6,29] and administered intramuscularly in the thigh contralateral to the ALVAC administration site. During DNA primes (weeks 0 and 4), thirteen animals received the DNA-SIV constructs in 1.4 mL PBS (Vaccine group), whereas twelve animals received the DNA-SIV constructs together with 250 µg of DNA-IGF-1 in a total volume of 1.4 mL PBS (Vaccine+IGF-1 group). Additionally, during each immunization (weeks 0, 4, 8, and 12), thirteen animals were injected with five doses of 1 mL/each PBS (Vaccine group), whereas twelve animals were injected with five doses of 0.08 mg/kg of Increlex^®^ diluted in PBS. Increlex^®^ and PBS were administered subcutaneously at 2 h intervals without sedation, with 3 doses delivered prior to (5, 3, and 1 h prior) and 2 doses following (1 and 3 h following) each immunization.

Five weeks following the last immunization (week 17), the immunized macaques (*n* = 25) and naïve control macaques (*n* = 27) were sedated and exposed to SIV_mac251_ as previously described [6]. Viral challenges were performed intravaginally once per week through administering 11 repeated, low doses of pathogenic virus. The SIV_mac251_ used for viral challenge was produced in macaque cells (QBI#305342b, Quality Biological, Gaithersburg, MD, USA) and administered in a final volume of 1 mL of virus diluted in RPMI 1640 (Gibco, Waltham, MA, USA) to reach a 4000 TCID_50_/mL concentration (concentration assessed in 221 rhesus cells).

### 2.2. Extracellular Vesicle Isolation from Plasma and Active RAS Measurement

EVs were isolated from plasma collected at baseline from twenty-six animals vaccinated with ALVAC-SIV and SIV gp120 adjuvanted in alum (gp120alum ALVAC-SIV group [5]), on plasma collected at baseline and days 1, 2, 6, and 8 from nine animals of the DNA-IGF-1 administration study, and on plasma collected at baseline and following the third and fifth administrations of PBS/Increlex^®^ from nine animals of the Increlex^®^ administration study.

Blood was drawn directly in EDTA-containing tubes (K2 EDTA BD Vacutainer, Becton, Dickinson and Company, Franklin Lakes, NJ, USA). Plasma was separated via centrifugation for 30 min at 900× *g* and 20 °C and then cryopreserved at −80 °C. The isolation of plasma EVs was conducted as previously described [6]. Briefly, following thawing in ice, 1 mL of plasma was centrifuged for 10 min at 300× *g* and 4 °C, then transferred to a new tube and centrifuged for 10 min at 2000× *g* and 4 °C. Supernatants were then centrifuged for 30 min at 10,000× *g* and 4 °C, transferred to ultracentrifuge tubes (cat. #326814, Beckman Coulter, Brea, CA, USA), diluted with PBS maintained at 4 °C, and centrifuged for 2 h at 25,000 rpm (around 46,000× *g*) and 4 °C. Following ultracentrifugation, pellets were resuspended in 350 µL of PBS and cryopreserved at −80 °C.

Active RAS in EVs was measured using an RAS activation ELISA assay kit (Catalog # 17-497, MilliporeSigma, Rockville, MD, USA) following the manufacturer’s instructions. Briefly, EVs were thawed in ice and diluted in cold PBS in order to reach a concentration of 1 µg/µL. The plate was pre-rinsed with wash buffer, loaded with positive and negative controls and 50 µg of EVs, and incubated at room temperature (RT) for 1 h. Following incubation, the plate was washed and 50 µL of primary antibody solution was added to each well and incubated for 1 h at RT. Following incubation, the plate was washed again and 50 µL of secondary antibody solution was added to each well and incubated for 1 h at room temperature. Following incubation, the plate was washed again, rinsed twice with wash buffer not containing Tween^®^ 20, and 50 µL of chemiluminescent substrate was added to each well and incubated at RT for 1 h. Ten minutes following substrate addition, the plate was read using a Vicor^TM^ X4 2030 Multilabel reader (Perkin Elmer, Inc., Waltham, MA, USA) following the Luminescence protocol.

### 2.3. DNA-IGF-1 Plasmid Construction and Production

The DNA-IGF-1 plasmid for immunization was generated through synthesizing the *macaca mulatta igf-1* gene (gene sequence obtained from the NCBI Reference Sequence NM_001260726.1. Link: https://www.ncbi.nlm.nih.gov/gene/?term=NM_001260726.1, accessed on 11 June 2015; Figure 1) and inserting it in the pVR1332 plasmid vector. First, the synthetic *macaca mulatta igf-1* gene was assembled from synthetic oligonucleotides and inserted into a pMA-T intermediate vector (GeneArt^®^, Thermo Fisher Scientific, Waltham, MA, USA). A Kozak sequence and two stop codons were inserted at the 5′ and 3′ ends, respectively. The gene was codon-optimized for macaques, and restriction sites for SacII and EcoRI restriction enzymes were inserted at the 5′ and 3′ ends, respectively. The intermediate plasmid containing the *igf-1* gene was purified from transformed bacteria, and the concentration was determined via UV spectroscopy. The final construct was verified through sequencing, and the sequence congruence within the insertion sites was 100%. Then, the *macaca mulatta igf-1* gene was introduced in the final pVR1332 plasmid vector using the SacII and EcoRI restriction enzymes. The final DNA-IGF-1 construct was verified through sequencing, and the sequence congruence within the insertion sites was 100%. The DNA-IGF-1 plasmid for immunization was amplified by Aldevron, LLC (Fargo, ND, USA). Macaque IGF-1 expression by DNA-IGF-1 was confirmed in the supernatants of transfected cells via Western blot and ELISA.

### 2.4. DNA-IGF-1 Transfection In Vitro

The transfection of 293T cells (cat# crl-3216, American Type Culture Collection [ATCC], Manassas, VA, USA) with DNA-IGF-1 was performed using LipoD293^TM^ DNA in vitro transfection reagent (catalog # SL100668, SignaGen Laboratories, Ijamsville, MD, USA) and following the manufacturer’s instructions. Briefly, 293T cells were cultured in DMEM (Gibco, Thermo Fisher Scientific Inc.) containing 10% fetal bovine serum (R&D systems Inc., Minneapolis, MN, USA) and 1× anti-anti (Gibco, Thermo Fisher Scientific Inc.) at 80% confluence. Before transfection, the mix of LipoD293-DNA complex was prepared using two different plasmids, the DNA-IGF-1 and a control plasmid, and through incubating it at room temperature (RT) for 15 min. Following incubation, 293T cells were transfected in 12-well plates, and 100 µL of LipoD293-DNA complex containing 1µg of DNA was added to each well. Following transfection, cells were incubated in an incubator at 37 °C, and supernatants were collected at baseline and 1 and 3 days following transfection. Supernatants were centrifuged at 4 °C and 300× *g* for 10 min to eliminate remaining cells and subsequently at 10,000× *g* for 30 min at 4 °C to remove any remaining cell debris, and then they were stored at −80 °C.

### 2.5. Western Blot

A Western blot assay was conducted on cell supernatants collected 1 day following the transfection of 293T cells with DNA-IGF-1 or a control DNA plasmid. The same amounts of total proteins for DNA-IGF-1 and control plasmid samples were assayed. Supernatants were thawed and then boiled for 5 min at 100 °C with 2× sample buffer with 10% β-mercaptoethanol. The denatured proteins were separated using SDS-PAGE (NuPAGE 4–12% Bis-Tris Protein Gels, Thermo Fisher Scientific) for approximately 2 h at 100 A and then transferred to a previously methanol-activated hydrophobic PVDF (Immobilon-P PVDF, Millipore Sigma, St. Louis, MO, USA) for 1 h 30 min at 140 mA. The membranes were incubated overnight at 4 °C with primary antibodies to IGF-1 (cat# ab9572, Abcam, Boston, MA, USA) at a 0.2 µg/mL concentration in PBS containing 0.1% Tween 20 and 0.25% milk. Membranes were washed in PBS 0.1% Tween and exposed to a horseradish peroxidase-conjugated goat secondary anti-Rabbit antibody (1:10,000; ab6721, Abcam). Proteins were visualized via chemiluminescence using a ChemiDoc Imaging System (Bio-Rad Laboratories, Hercules, CA, USA). Densitometric analysis was performed using Image J open access software JS (V0.5.7). The quantifications of Western blot bands were measured using Image J open access software JS (V0.5.7). Briefly, a frame was created around the band of interest creating what is called a region of interest. An equal-size region of interest was created at the same molecular weight of the IGF-1 protein in the negative control lane. Background regions of interest were created below the regions of interest. Next, the pixel densities inside the four frames were measured and inverted using the formula 255-X (X = value recorded for each region of interest/background region of interest). The final net densities were calculated through subtracting the inverted intensity of the region of interest from the inverted density of the background region of interest. See the data availability statement for the uncropped blot and densitometry analysis.

### 2.6. Free IGF-1 ELISA

Free IGF-1 levels were measured in plasma and supernatants of cells transfected with DNA-IGF-1 via human free IGF-1 immunoassay (Catalog # DFG100, bio-Techne, R&D systems Inc., Minneapolis, MN, USA) following the manufacturer’s instructions. Briefly, plasma and supernatants were thawed in ice; the plasma was then tested undiluted, whereas supernatants were tested undiluted and diluted 1:5. The ELISA plate was loaded with 20 µL/well of free IGF-I biotinylated antibody, followed by 50 µL of standards, controls, and plasma samples, and incubated at RT for 1 h. Following incubation, the plate was washed, and 100 µL of diluted free IGF-1 conjugate solution was added to each well and incubated at RT for 30 min. Following incubation, the plate was washed again. Each well was filled with 100 µL of substrate solution, incubated at RT for 10 min, and 100 µL of stop solution was added. The plate was read using a Vicor^TM^ X4 2030 Multilabel reader following a protocol of absorbance at 450 nm. The optical density (OD) was used to generate a four-parameter logistic (4-PL) curve fit using the standards and then to calculate the free IGF-1 concentration (ng/mL) in plasma samples.

### 2.7. Total IGF-1 ELISA

Plasma total IGF-1 was measured via human IGF-1 immunoassay (Catalog # DG100, bio-Techne, R&D systems Inc.) following the manufacturer’s instructions. First, plasma was pretreated in order to release the protein-bound form of the IGF-1. Briefly, after thawing in ice, 20 µL of plasma was added to 380 µL of pretreatment A solution, vortexed, and incubated at room temperature for 10 min. Following incubation, 50 µL of mix was added to 200 µL of pretreatment B solution and mixed. Treated plasma was then assayed either undiluted or diluted 1:2 or 1:4 with calibrator diluent. The ELISA plate was loaded with 150 µL/well of assay diluent, 50 µL of standards, controls, and treated samples, and incubated at 4 °C for 2 h. Following incubation, the plate was washed and 200 µL of cold human IGF-1 conjugate was added to each well and incubated at 4 °C for 1 h. Following incubation, the plate was washed again, and 200 µL of substrate solution was added to each well and incubated at RT for 30 min. Following incubation, 50 µL of stop solution was added to each well and the plate was read using a Vicor^TM^ X4 2030 Multilabel reader following the protocol of absorbance at 450 nm. The OD was used to generate a log/log curve fit using the standards and then to calculate the total IGF-1 concentration (ng/mL) in plasma samples.

### 2.8. IgG Serum Titers to gp120

Total gp120 IgG antibodies were measured via ELISA as previously described [30]. Briefly, ELISA plates were coated overnight at 4 °C with 50 ng of SIV_mac251-M766_ gp120 protein/well in 100 µL of 50 mM sodium bicarbonate buffer (pH 9.6). The plates were blocked with Superblock (Thermo Fisher Scientific) for 1 h at RT, and, following incubation, 100 µL of serially diluted serum samples with sample diluent (Avioq, Durham, NC, USA), was added to the wells. Plates were incubated for 1 h at 37 °C, washed, and incubated for 1 h at 37 °C with 100 µL of anti-human HRP (diluted at 1:120,000). The plates were next washed and developed using K-Blue Aqueous substrate and through stopping the reaction with 2N Sulfuric acid. Plates were acquired using a Molecular Devices E-max plate reader at 450 nm. Titers were calculated as the higher dilution that provided an OD value double that of the average normal rhesus.

### 2.9. Measurement of IgG Subclass Antibodies

Concentrations of SIV gp120-specific IgG1, IgG2, IgG3, and IgG4 antibodies were measured in plasma collected 2 weeks following the last immunization via ELISA. Ten rows of a 96-well Immulon 4 microtiter plate (VWR) were coated overnight at 4 °C with 50 ng gp120 per well in PBS at pH 7.2. To generate a standard curve with a known concentration of IgG1, IgG2, IgG3, or IgG4, 2 rows of the plate were coated with 50 ng per well recombinant CD40 (NHP Reagent Resource). The following day, the plate was washed with PBST and blocked with PBST containing 0.1% BSA (ELISA buffer; EB). Plasma samples diluted in EB were then added to gp120 wells. Duplicate dilutions of anti-CD40 rhesus IgG1, IgG2, IgG3, or IgG4 antibody (NHP Reagent Resource) were added to the CD40 wells. Following overnight reaction at 4 °C, the plate was washed and treated for 1 h at 37 °C with 0.5 µg/mL of one of the following biotinylated monoclonal antibodies (NHP Reagent Resource) diluted in EB: anti-rhesus IgG1 clone 7H11 (ena), anti-rhesus IgG2 clone 8D11 (dio), anti-rhesus IgG3 clone 6F5 (tria), or anti-rhesus IgG4 clone 7A8 (tessera). Plates were washed, treated with 1/4000 neutralite avidin peroxidase (SouthernBiotech, Homewood, AL, USA) for 30 min at room temperature, and developed with TMB substrate and H_2_SO_4_ stop solution. After recording absorbance at 450 nm, a standard curve constructed from the anti-CD40 IgG subclass antibody was used to interpolate the concentration of anti-gp120 IgG of the same subclass in plasma.

### 2.10. Measurement of Plasma IgA Antibodies

Concentrations of SIV gp120-specific IgA in plasma were measured via ELISA using previously described methods [31]. Briefly, microtiter plates were coated overnight with 50 ng of gp120 per well. The following day, plates were blocked with EB and loaded with dilutions of plasma and standard. The plasma was depleted of IgG using Protein G Sepharose (PG; GE Healthcare, Chicago, IL, USA) since the secondary antibody was found to cross-react with rhesus IgG. The standard was IgG-depleted pooled serum from SIV-vaccinated/infected macaques [31]. Plates were treated the following day with goat IgG containing anti-monkey IgA that had been purified from antiserum (Novus Biologicals, Centennial, CO, USA) in the laboratory using PG and then biotinylated using the Pierce Antibody Biotinylation Kit as instructed by the manufacturer. The plates were then developed with neutralite avidin peroxidase and TMB as above.

### 2.11. Serum Neutralizing Antibodies

Neutralizing antibodies levels were assessed in the serum of immunized macaques collected at baseline and week 14, through an assay of reduction in luciferase reporter gene expression [6]. Serum was diluted and incubated with different viruses (200 TCID_50_) for 1 h at 37 °C. SIV_mac251_._6_ (ID #1636DB2), SIV_smE660/BR-CG7G_._IR1_ (ID #1370DB2), SIV_smE660/BR-CG7G_._IR1_ (ID #1634DB2), and SIV_mac251_ (challenge virus) viruses were tested. Following incubation, TZM-bl cells in growth medium + 20 μg/mL DEAE dextran were added to each well. Concomitantly, wells containing cells and viruses, or only cells, were used as the virus control and background control, respectively. Following 48 h incubation, the cells were lysed, the luminescence was measured, and the background relative luminescence was subtracted. The sera dilution in which the relative luminescence units were reduced by 50% or 80% compared to that in the virus control was defined as the neutralization titer for each sample.

### 2.12. Pepscan

Serum samples were assayed via PEPSCAN analysis using SIV_mac251_ gp120 linear peptides as previously described [30]. ELISA plates were coated through incubating them overnight at 4 °C with 1000 ng each of 89 overlapping peptides encompassing the whole sequence of the SIV_mac251_ gp120 protein. Following incubation, plates were blocked with blocking buffer, then given 100 µL of serum (diluted at 1:50) and incubated again for 1 h at 37 °C. Plates were then washed, supplied with 100 µL HRP-labeled anti-human IgG (1:120,000 dilution), and incubated for 1 h at 37 °C. Plates were washed again and developed using K-Blue Aqueous substrate (Neogen, Lansing, MI, USA) for 30 min at RT. The reaction was stopped with 100 µL 2 N Sulfuric acid/well, and the plate was read with a Molecular Devices E-max plate reader at 450 nm. The OD for each sample was used for the analysis.

### 2.13. SIV-Specific Plasma and Mucosal IgG-Binding by Antibody Multiplex Assay

Plasma, rectal, and vaginal mucosal env-SIV IgG were measured from plasma, rectal, and vaginal mucosa swabs collected at baseline and 2 weeks following the last vaccination (week 14). Plasma and secretions were analyzed through custom SIV binding antibody multiplex assays (SIV-BAMA) as previously described [30,32]. Mucosal swabs were processed, examined for blood contamination, and measured for a semiquantitative evaluation of hemoglobin. The concentrations of mucosal total IgG were evaluated via a custom macaque total IgG ELISA using purified IgG (DBM5) from an SIV-infected macaque [33] as a positive control to calculate SIV antibody concentration. Antibodies against native V1/V2 epitopes were quantified through binding assays against scaffolded SIV V1/V2 antigens expressed as gp70 fusion proteins related to the CaseA2 antigen used in the RV144 correlate study (provided by A. Pinter, New Jersey Medical School, Newark, NJ, USA). These proteins contained the glycosylated, disulfide-bonded V1/V2 regions of SIV_mac239_, SIV_mac251_, and SIV_smE660_ (corresponding to AA 120–204 of HXB2 Env), fused to residue 263 of the Fr-MuLV SU (gp70) protein. The positive control for each antigen was tracked via Levey–Jennings charts. For plasma, the binding magnitude is reported as the median fluorescence intensity (MFI) at 1:2000 dilution, whereas, for mucosal secretions, the binding magnitude is reported as specific activity, calculated as MFI × dilution ÷ total IgG (concentration in µg/mL).

### 2.14. ADCC against SIV_mac251_-Infected Cells

The ADCC-Luc assay was used to measure the ADCC activity directed against SIV_mac251_-infected cells. The assay was conducted as previously described [6]. Briefly, an infectious molecular clone of SIV_mac251_ virus that encodes Renilla luciferase was used to infect CEM.NKRCCR5 target cells (NIH AIDS Research and Reference Reagent Program). Target cells were then cocultured with effector cells (ratio of effector/target of 30:1) in the presence of different dilutions of plasma. Following incubation, the change in relative light units (RLU; Vivi Ren luciferase assay; Promega, Madison, WI, USA) due to the loss of target cells was used to calculate the percent specific killing. The following formula was used: (number of RLU of target + effector well number of RLU of test well)/number of RLU of target + effector well. Finally, the activity measured in pre-vaccination samples was subtracted from post-vaccination samples to calculate the adjusted percentages of specific ADCC killing.

### 2.15. V2-Specific ADCC Killing

V2-specific antibody-dependent cellular cytotoxicity (ADCC) activity was measured as previously described [30]. Briefly, EGFP-CEM-NKr-CCR5-SNAP target cells (kindly provided by Dr. G. K. Lewis) were coated with gp120 protein, labeled with SNAP-Surface^®^ Alexa Fluor^®^ 647 (New England Biolabs, Ipswich, MA, USA), and incubated with purified F(ab′)_2_ fragments from NCI09 or NCI05 monoclonal antibodies. Target cells were then incubated with 1:100 diluted heat-inactivated plasma samples collected from vaccinated macaques. Following incubation, human PBMC effector cells were added to target cells at a ratio of effector/target (E/T) cells of 50:1 and incubated for 2 h at 37 °C. Following incubation, cells were analyzed via flow cytometry. Normalized percent killing was calculated using the formula: (killing in the presence of plasma or plasma + F(ab′)_2_ − background)/(killing in the presence of positive control − background) × 100. The V2-specific ADCC killing was calculated using the formula: ADCC killing measured in the absence of F(ab′)_2_ − ADCC killing measured in the presence of NCI09 or NCI05 F(ab′)_2_.

### 2.16. Flow Cytometry Analysis

#### 2.16.1. Monocytic Myeloid Cells

Monocytic myeloid cells were assessed via flow cytometry. Briefly, cryopreserved PBMCs (5–10 × 10^6^ cells) collected at the end of vaccination (week 13) were thawed and incubated with Fluorochrome-conjugated mAbs. The following antibodies were used: PE-Cy7 anti-CD3 (clone SP34-2; cat. #563916, 2.0 µL), PE-Cy7 anti-CD20 (clone 2H7; cat. #560735, 1.0 µL), BV786 anti-NHP-CD45 (clone D058-1283; cat. #563861, 3.0 µL), APC anti-CD14 (clone M5E2; cat. #561390, 7.5 µL), FITC anti-CD16 (clone 3G8; cat. #555406, 5.0 µL), BV421 anti-CD192 (CCR2; clone 48607; cat. #564067, 3.0 µL), PE-CF594 anti-CD184 (CXCR4; clone 12G5; cat. #562389, 5.0 µL), all from BD Biosciences, and HLA-DR-APC-Cy7 (clone L243; cat. #307618, 5.0 µL) from BioLegend (San Diego, CA, USA). Dead cells were excluded using Aqua LIVE/DEAD viability dye (cat. #L34957, 1 µL; Thermo Fisher Scientific, Waltham, MA, USA). Myeloid cells were identified as CD45^+^ Lineage negative (CD3 and CD20), while total monocytes were identified as Lin^−^CD45^+^HLA-DR^+^. Monocytes were then categorized as classical (CD14^+^CD16^−^), intermediate (CD14^+^CD16^+^), and non-classical (CD14^−^CD16^+^) according to the expression of CD14 and CD16 [29].

#### 2.16.2. Innate Lymphoid Cells in Rectal Mucosa

NK cells were measured in rectal mucosa before vaccination and 1 week following the ALVAC and ALVAC+gp120 immunizations (week 9 and 13). Rectal biopsies were used for the isolation of mononuclear cells, and 3 × 10^6^ cells (if available) were used for functional characterization. Cells were incubated for 12 h in RPMI 1640 with 10% fetal bovine serum and Antibiotic-Antimitotic solution (Sigma-Aldrich, Saint Louis, MO, USA; R10) in the presence of 50 µg/mL of Gentamicin (Gibco), 3 µg/mL of Brefeldin A, and 2 µM of Monesin (Gibco). During incubation, cells were left unstimulated, or stimulation was accomplished with an *env* peptide pool (final concentration 2 µg/mL) or a mix of 50 ng/mL of Phorbol 12-myristate 13-acetate (PMA) and 750 ng/mL of Ionomycin (Sigma-Aldrich). The *env* peptide pool was constituted by 20 amino acid peptides encompassing the whole SIV_mac251_ gp120 and overlapping 15 amino acids. Flow cytometry staining to identify cellular surface or intracellular markers was performed with anti-human fluorochrome-conjugated monoclonal antibodies (mAbs), which cross-react with rhesus macaque cells. The following mAbs were used: FITC anti-CD45 (clone D058-1283; cat. #557803, 4 µL); APC-Cy7 anti-CD3 (clone SP34-2; cat. #557757, 3 µL); V450 anti-IFN-γ (clone B27; cat. #560371, 2 µL) from BD Biosciences (San Jose, CA, USA); eVolve655 anti-CD20 (clone 2H7; cat#86-0209-42, 4 µL); PerCP-eFluor710 anti-IL-22 (clone IL22JOP; cat. #46-7222-82, 3 µL); PE-Cyanine7 anti-IL-17 (clone eBio64DEC17; cat. #25-7179-42, 3 µL) from eBioscience (Thermo Fisher Scientific); BV785 anti-CD14 (clone M5E2; cat. #301840, 1 µL); APC anti-NKp44 (clone P44-8; cat. #325110, 3 µL); BV605 anti-TNF-α (clone Mab11; cat. #502936, 3 µL) from BioLegend; PE anti-NKG2A (clone Z199; cat. #IM3291U, 3 µL) from Beckman Coulter. Dead cells were excluded using aqua LIVE/DEAD viability dye (cat. #L34957, 1 µL; Thermo Fisher Scientific). For intracellular staining, the cells were permeabilized and fixed using the BD Cytofix/Cytoperm kit (BD Biosciences). Mucosal NK cells were identified via gating on live CD45^+^ lymphocytes Lineage negative (CD20, CD14 and CD3) and classified depending on their expression of the surface markers NKG2A and NKp44 as NKG2A^+^, NKp44^+^, or NKG2A^−^NKp44^−^ innate lymphoid cells as previously described [5].

#### 2.16.3. Systemic and Mucosal CD4^+^ T-Cells

The frequencies of CD4^+^ T-cell subsets were assessed in blood and rectal mucosa in week 13. Systemic CD4^+^ T-cells were analyzed in cryopreserved PBMCs (5–10 × 10^6^ cells), whereas mucosal CD4^+^ T-cells were analyzed in mononuclear cells isolated from rectal biopsies. PBMCs and mucosal cells were stained with PerCPCy5.5 anti-CD4 (clone L200; cat. #552838, 5 μL), AlexaFluor 700 anti-CD3 (clone SP34-2, cat. #557917, 0.2 mg/mL), BV650 anti-CCR5 (clone 3A9, 5 μL), PeCy5 anti-CD95 (clone DX2, #559773, 5 μL), and FITC anti-Ki67 (cat. #556026, 10 μL) from BD Biosciences; PE-eFluor 610 anti-CD185 (CXCR5; clone MU5UBEE, #61-9185-42, 5 μL) from eBioscience; APC Cy7 anti-CXCR3 (clone G025H7, cat. #353721, 5 μL) and BV605 anti-CCR6 (clone G034E3, cat. #353419, 5 μL) from BioLegend; PE anti-CD38 (clone OKT10; PR-3802) and APC anti-α_4_β_7_ (clone A4B7; PR-1421), provided by the NIH Nonhuman Primate Reagent Resource (R24 OD010976, and NIAID contract HHSN 272201300031C). Dead cells were excluded using Violet LIVE/DEAD viability dye (cat. #L34964, 1 µL; Thermo Fisher Scientific). Vaccine-induced CD4^+^ T-cell identification was performed via gating on live CD3^+^CD4^+^Ki67^+^. The expression of CXCR3 and CCR6 receptors was used to identify Th1 or Th2 populations as previously described [29]. LSRII and FACSDiva (v9.0) software (BD Biosciences) were used to perform flow cytometry acquisitions through acquiring a minimum of 500,000 events. FlowJo v 10.1 (TreeStar, Inc., Ashland, OR, USA) was used for data analysis.

### 2.17. Viral RNA, Proviral DNA and CD20^+^/CD4^+^/CD8^+^ Cells Counts

Plasma collected from animals was analyzed to quantify the RNA copies of SIV_mac251_. Quantification was performed through nucleic acid sequence-based amplification as previously described [6,29]. The proviral DNA load in vaginal mucosa was analyzed through following a previously described protocol [6]. Counts of CD20^+^, CD4^+^, and CD8^+^ T-cell in whole blood were assessed via flow cytometry and following a previously described method [29].

### 2.18. Intracellular Cytokines of Macaque Blood NK Cells Using IGF-1 and gp120 Stimulation

The levels of NKG2A^+^ NK cells and cytokine expression by these cells were measured in PBMCs collected 2 weeks following vaccination from 8 vaccinated macaques with wild-type DNA/ALVAC/gp120/alum vaccine [30] and stimulated in the presence or absence of IGF-1. Briefly, PBMCs collected from rhesus macaques were thawed and cultured in R10 in the presence or absence of IGF-1 (Increlex^®^, 150 ng/mL) and left unstimulated or stimulated with gp120 pooled peptides or PMA/Ionomycin in an incubator at 37 °C for 2 h. Subsequently, Golgi plug and Golgi stop were added to the culture and incubated for 16 additional hours. Next, the cells were stained after incubation. Dead cells were excluded using Live/Dead Blue dye (cat. no. L34962, 0.5 μL) from Thermo Fisher. Surface staining was performed with the following antibodies for 30 min at RT: BV786 anti-CD45 (D058-1283; cat. no. 563861, 5 μL), Alexa 700 anti-CD3 (SP34-2; cat. no. 557917, 5 μL), Alexa 700 anti-CD20 (2H7; cat. no. 560631, 5 μL), and BV711 anti-CD8 (RPA-T8; cat. no. 563677, 5 μL) from BD Biosciences; APC-H7 anti-CD11b (ICRF44; cat. no. 47-0118-42, 5 μL) from eBioscience; PE-Cy7 anti-NKG2A (Z199; cat. no. B10246, 5 μL) from Beckman Coulter. Following incubation, cells were washed and permeabilized using Foxp3/transcription buffer set (cat. no. 00-5523-00) from eBioscience and following the manufacturer’s instructions. Subsequent intracellular staining was conducted with the following antibodies for 30 min at RT: BV750 anti-TNF-α (MAb11; cat. no. 566359, 5 μL), BUV396 anti-IFN-γ (B27; cat. no. 563563, 5 μL), and BV510 anti-GranB (GB11; cat. no. 563388, 5 μL) from BD Biosciences; and FITC anti-Perforin (pf-344; cat. no. 3465-7, 5 μL) from Mabtech (Cincinnati, OH, USA). Flow cytometry acquisitions were performed on a FACSymphony A5 and examined using FACSDiva software (BD Biosciences). NKG2A^+^ NK cells were gated as singlets/live cells/CD45^+^/CD3^−^CD20^−^CD11b^−^/NKG2A^+^CD8^+^ cells. The frequencies of cytokine^+^ NKG2A^+^NK cells were determined as frequencies of cells in CD45^+^ cells.

### 2.19. Efferocytosis with IGF-1 Stimulation

The effect of IGF-1 stimulation on the frequency of CD14^+^ efferocytes and their engulfing capability was investigated with the Efferocytosis Assay kit (cat. #601770, Cayman Chemical Company, Ann Arbor, MI, USA) following a previously described procedure [6] with minor modifications. Due to the lack of availability of cells collected from vaccinated animals treated with IGF-1 or not, the efferocytosis assay was conducted on CD14^+^ cells collected from 9 naïve animals stimulated in vitro with or without IGF-1.

#### 2.19.1. Effector Cells

CD14^+^ cells were isolated from cryopreserved PBMCs (10 × 10^6^ cells) using non-human primate CD14 MicroBeads (#130-091-097) and AutoMACSpro (Miltenyi Biotec, Gaithersburg, MD, USA) and following manufacturer instructions. At the end of the separation, cells were counted and stained with CytoTell^TM^ Blue provided in the Efferocytosis Assay kit, following manufacturer instructions, as previously described [6].

#### 2.19.2. Target Cells

Neutrophils collected from one unrelated macaque were used as target cells. Neutrophils were isolated in a multistep protocol including Ficoll Plaque (GE Healthcare) and subsequent incubation with Dextran, and then stained with CFSE, as previously described [6]. Following CFSE staining, the neutrophils were treated with Staurosporine to induce apoptosis as indicated by the Efferocytosis Assay kit.

#### 2.19.3. CD14^+^ Cells and Neutrophils Coculture

The coculture of effector and apoptotic target cells was performed at a ratio 1:3 of effector and target cells, through culturing cells in R10 in the presence or absence of IGF-1 (Increlex^®^, final concentration 150 ng/mL). Cells were incubated overnight in an incubator at 37 °C. Effector and target cells cultured alone were used as controls. At the end of the incubation, cells were washed, fixed, and analyzed via flow cytometry. Acquisitions were performed with FACSymphony A5 and FACSDiva software (BD Biosciences). FlowJo v 10.1 (TreeStar, Inc., Ashland, OR, USA) was used for data analysis. The percentage of CD14^+^ efferocytes was assessed as the percentage of CFSE^+^ cells (neutrophils) in the CytoTell^TM^ Blue^+^ cells (CD14^+^ cells). The engulfing capacity was determined as the median fluorescence intensity of CFSE in CFSE^+^ CytoTell^TM^ Blue^+^ cells (CFSE^+^ CD14^+^ cells).

### 2.20. Trogocytosis

Trogocytosis was measured using a previously described assay [34] conducted with plasma collected at baseline and after the end of vaccination (week 14). Briefly, CEM.NKR.CCR5 cells were washed with PBS and stained at RT for 5 min with 2 μM PKH26 (Sigma-Aldrich) in Diluent C. Following incubation, cells were washed, resuspended in R-10, and incubated for 1 h at RT with WT gp120. Cells were washed again with R-10 and incubated with 300-fold-diluted plasma samples. Cryopreserved healthy control PBMCs were next thawed and added in R-10 at an effector to target (E:T) cell ratio of 50:1 and then incubated for 5 h at 37 °C. Following incubation, cells were washed, stained with live/dead aqua fixable stain and anti-CD14 APC-H7-conjugated antibody (clone MΦP9, BD), washed again, and fixed with 4% formaldehyde (Tousimis, Rockville, MD, USA). Fluorescence was evaluated on an LSRII flow cytometer (BD Biosciences). The trogocytosis score was evaluated through measuring the percentage of the live CD14^+^ cells positive for PKH26.

### 2.21. Statistical Analysis

Statistics were calculated using GraphPad Prism. The statistical analyses were performed as two-tailed. All analyses were considered exploratory analyses, and *p*-values were not corrected for multiple comparison. The Mann–Whitney–Wilcoxon test was used to compare continuous factors between vaccinated and vaccinated+IGF-1 macaques. All scatter plots report the mean and standard deviation (SD), except where stated otherwise. Spearman’s rank correlation was used to infer linear relationships between measured variables. For correlations including the numbers of challenges required for infection, 12 was assigned as the value of the right-censored numbers for any value greater than 11. Following exposure of the animals to SIV_mac251_, the viral acquisitions of different groups were compared using the exact log-rank Mantel–Cox test of the discrete-time proportional hazards model.

## 3. Results

### 3.1. Administration of DNA Encoding IGF-1 and Recombinant IGF-1 Protein Increlex^®^ to Macaques

In a previous study, we identified 12 genes involved in RAS pathway activation that were associated with vaccine efficacy [5]. RAS in its active-GTP bound state can be detected in plasma extracellular vesicles (EVs) [35]. EVs can carry immunomodulatory molecules, influencing the immune response and facilitating communication between immune cells during infection, inflammation, and immune system regulation [36,37]. We therefore isolated EVs from cryopreserved plasma from the same study using multiple-step centrifugation and ultracentrifugation [6] and found that active RAS levels in the EVs prior to vaccination correlated with a decreased risk of SIV_mac251_ acquisition (ρ = 0.48, *p* = 0.0121; Figure 2a), confirming transcriptome analyses and validating the role of RAS activation in vaccine protection. These data prompted us to formulate the hypothesis that an RAS-activating agent could act as an adjuvant in vaccine immunogenicity and improve vaccine efficacy. To test this, we generated a plasmid DNA encoding macaque IGF-1 based on the *macaca mulatta igf-1* gene sequence obtained from the NCBI Reference Sequence NM_001260726.1 (https://www.ncbi.nlm.nih.gov/gene/?term=NM_001260726.1, accessed on 11 June 2015), which represents the unspliced mRNA. The comparison between human and macaque *igf-1* genes revealed a single nucleotide mutation at position 44, reflecting an amino acid change from Methionine (M) in humans to Isoleucine (I) in macaques (Figure 1). Macaque IGF-1 expression was confirmed in the supernatants of transfected cells via Western blot at 24 h (Figure 2b) and via ELISA at 24 and 72 h after transfection (*p* = 0.002 at both intervals, compared to DNA transfected controls; Figure 2c).

Next, we performed a dose–response study through administering 250, 500, and 1000 μg of DNA-IGF-1 intramuscularly to three groups of three macaques each (nine animals total; Figure 2d). Blood was collected prior to administration and at several timepoints (1, 3, 6, 8, 10, 13, and 15 days) following the administration, and the content of total and free IGF-1, as well as RAS activation, were evaluated. The total and free IGF-1 plasma levels did not change significantly in the three groups except transiently at day 6, when the three groups differed in their total IGF-1 (*p* = 0.025; Figure 2e) and free IGF-1 (*p* = 0.0036; Figure 2f) levels. However, these differences may be due to normal fluctuation in IGF-1 levels, as reported in humans [38], rather than a consequence of the administration of different amounts of DNA-IGF-1. Following the isolation of plasma EVs and the evaluation of active RAS levels, no significant changes in RAS activation were identified in the three groups (Figure 2g). We next tested the ability of the FDA-approved recombinant human IGF-1 protein, Increlex^®^, to induce RAS activation in macaques in vivo. We treated five macaques with five injections of 80 μg/kg each of Increlex^®^ and four macaques with saline at the same timepoints, as controls (Figure 2h). Each injection was subcutaneously administered at 2 h intervals, and blood was collected before the first inoculation, 1 h following the third and fifth injections (5 and 9 h, respectively). As expected, the levels of total (*p* = 0.063 and 0.032, Figure 2i) and free (*p* = 0.016; Figure 2j) IGF-1 in plasma were significantly higher in the Increlex^®^-treated animals compared to the saline controls. Additionally, active RAS levels measured in plasma EVs trended higher in Increlex^®^-treated animals than in controls (*p* = 0.063; Figure 2k).

### 3.2. Coadministration of IGF-1 with DNA/ALVAC/gp120/alum Vaccine

Since Increlex^®^ administration in macaques caused a transient increase in both IGF-1 levels and RAS activation, we next designed a study to assess its potential adjuvant proprieties. We randomized 51 female rhesus macaques into three groups (Figure 3a). Thirteen animals (Vaccine group) were primed twice with virus-like particles (VLPs) at weeks 0 and 4 delivered through the coadministration of DNA-SIV *gag* and *env*, boosted once with ALVAC-SIV, which also produces VLPs, and once again with ALVAC-SIV in association with SIV_mac251_ and SIV_smE660_ gp120 proteins adjuvanted in alum, administered in the contralateral thigh. Each immunization was administered intramuscularly at 4-week intervals. A second group of twelve animals (Vaccine+IGF-1 group) was identically immunized at the same time points, but with the addition of 250 μg of DNA-IGF-1 to the two DNA-SIV primes and a total of 20 doses (5 at each immunization) of 80 μg/kg Increlex^®^ administered at 2 h intervals (3 prior to and 2 following each immunization: −5, −3, −1, +1, and +3 h; Figure 3a). The remaining 27 monkeys were left naïve in a control group (Figure 3a). As expected, the free IGF-1 levels in plasma were higher in animals treated with DNA-IGF-1 and Increlex^®^ during each immunization than in the animals treated with PBS (*p* < 0.0001 at each timepoint; Figure 3b).

### 3.3. Modulation of the Antibody Response to Vaccination by IGF-1 Treatment

The adjuvant effect of IGF-1 on the DNA/ALVAC/gp120/Alum vaccine immunogenicity was investigated in plasma, PBMCs, and mucosal samples collected prior and following immunizations. At first, we investigated the ability of IGF-1 to modify the antibody responses elicited by vaccination both at the systemic and mucosal levels. Two weeks following the last immunization (week 14), the titers of IgG binding antibodies to the whole SIV_mac251_ gp120 were similar in macaques vaccinated with or without IGF-1 treatments (Figure 3c), and, similarly, the IgA, IgG1, IgG2, IgG3, and IgG4 levels of the whole SIV_mac251_ gp120 did not differ between the two groups (Appendix A). Next, we measured the serum levels of neutralizing antibodies recognizing SIV viruses with different sensitivity to antibody-mediated neutralizations (tiers). Two weeks following the last immunization (week 14), the coadministration of IGF-1 diminished the levels of neutralizing antibodies recognizing the tier-1A SIV_mac251_._6_ virus (*p* = 0.026; Figure 3d), while the levels of neutralizing antibodies recognizing tier-1A and -B SIV_smE660_ or the SIV_mac251_ challenge viruses were similar in the two groups (Appendix A).

To further characterize the antibody responses targeting the variable and constant regions of gp120, we performed Pepscan ELISA using overlapping 15-mer peptides encompassing the whole sequence of SIV_mac251_-gp120. The analysis revealed no differences in the overall responses against the five constant (C1–C5) regions of gp120 between the vaccine and vaccine+IGF-1 groups (Figure 3e).

In human clinical trials and in our previous macaque studies, the antibody responses to the V1/V2 region were associated with the risk of HIV and SIV acquisition [5,39]. Macaque studies demonstrated an increased risk associated with the level of antibody response to V1 [30]. Therefore, we investigated in detail if IGF-1 affected plasma antibody responses to linear peptides encompassing the V1 and V2 regions of SIV_mac251_ (Appendix A). We found that the Vaccine+IGF-1 group elicited significantly lower antibody responses to V1 peptide 17 (*p* = 0.043; Figure 3f) but higher response to peptide 25 (*p* = 0.026; Figure 3g), which partially overlaps with peptide 24 of the V1a region of gp120. Notably, antibody responses targeting V1a were found to correlate with an increased risk of virus acquisition in prior macaque studies [30]. 

Next, we evaluated the effect of IGF-1 on the induction of systemic and mucosal antibodies recognizing conformational epitopes through measuring the antibody responses recognizing an array of different V1/V2 scaffolds and whole gp120 proteins, as well as gp41 and p55-gag proteins. Two weeks following the last immunization, the IgG Ab responses to both gp70-V1/V2 of SIV_mac239_ and SIV_mac251WY30_ were higher in the rectal secretions (*p* = 0.047 and *p* = 0.062) of animals treated with IGF-1 (Figure 3h and Appendix A). In the IGF-1 group, the responses to gp70-V1/V2 of SIV_smE660_ were higher in both rectal and vaginal secretions (*p* = 0.010 and 0.021; Figure 3h and Appendix A).

### 3.4. IGF-1 Treatment Reduces V2-Specific ADCC and Transiently Increases the Frequency of Mucosal NKp44^+^ Cells

Since, in prior studies, we observed that V2-specific antibody-dependent cellular cytotoxicity (ADCC) was associated with decreased risk of SIV acquisition [6,30], we investigated whether IGF-1 affected ADCC against SIV-infected cells or target cells coated with SIV gp120 protein. In plasma collected 2 week following the last immunization, the total ADCC measured against SIV_mac251_-infected cells did not differ between the Vaccine and Vaccine+IGF-1 groups (Figure 4a). Similarly, total ADCC measured against target cells coated either with V1-deleted or wild-type SIV_mac251_-gp120 proteins in plasma collected 5 weeks following the last immunization did not show any difference between the two groups (Appendix A). Surprisingly, however, IGF-1 induced a lower level of V2-specific ADCC, identified through the inhibition of the assay with mAb NCI05 Fab’ (*p* = 0.024; Figure 4b). Interestingly, a lesser decrease than with NCI05 Fab’ was observed using NCI09 Fab’ (*p* = 0.089; Figure 4c). The mAbs NCI05 and NCI09, respectively, target the V2 coil-helical and b-sheet conformations, and the passive transfer of NCI05 but not NCI09 is able to delay SIV_mac251_ acquisition [40].

We next investigated the innate lymphoid cells (ILCs) present in the rectal mucosa to understand if IGF-1 could affect their expression of NKG2A and NKp44 receptors. In previous studies, the frequency of vaccine-induced rectal NKp44^+^ ILCs that produce IL-17 following stimulation with peptides encompassing the env sequence of SIV was correlated with reduced risk of SIV acquisition [5]. The IGF-1 treatment of vaccinated macaques resulted in a transient increase in the frequency of all NKp44^+^ ILCs measured 1 week following the ALVAC-SIV boost (week 9; *p* = 0.011; Figure 4d).

### 3.5. IGF-1 Treatment Alters Blood Monocyte Subsets Frequency, Decreases Trogocytosis and CCR5 Expression on Activated CD4^+^ T-Cells

In prior studies conducted in male and female macaques, the ALVAC-based vaccine-induced engagement of CD14^+^ classical monocytes was a strong correlate of decreased risk of SIV acquisition [6,29]. Moreover, RAS activation in monocytes can be important to activate pro-inflammatory responses [41], suggesting that IGF-1 could affect the frequencies of monocyte subsets and their functions. Therefore, we investigated the systemic monocytes in PBMCs collected 1 week following the last immunization (week 13) via flow cytometry. Following vaccination, the overall frequency of total monocytes was higher in Vaccinated+IGF-1 than in Vaccinated-only animals (*p* < 0.0001; Figure 4e). In animals administered Vaccine+IGF-1, the frequency of classical monocytes (CD14^+^CD16^−^) was higher than in the Vaccine group (*p* = 0.009; Figure 4f). As expected, the frequencies of intermediate (CD14^+^CD16^+^) and non-classical (CD14^−^CD16^+^) monocytes were also lower in the Vaccine+IGF-1 group (*p* < 0.0001 and *p* = 0.0006, respectively; Figure 4g and Appendix A). Additionally, the analysis of trogocytosis mediated by the sera of immunized animals demonstrated that IGF-1 administration with vaccination significantly decreases this response (Figure 4h).

Prior work indicated that the induction of CD4^+^ T-cells through vaccination is a correlate of the risk of HIV/SIV acquisition in humans and in macaques [29,42]. More particularly, vaccine-induced α_4_β_7_^+^CD4^+^ T-cells expressing low or no CCR5 correlated with a decreased risk in a prior macaque study [29].

Because the gut-homing CCR5^+^CD4^+^ cells that express the integrin receptor α_4_β_7_ are a potential target for HIV/SIV [43], their induction could fuel infection and decrease vaccine efficacy. Therefore, we investigated the ability of IGF-1 to modify the frequency of these cells following vaccination in the presence or the absence of IGF-1 at systemic and mucosal sites. Strikingly, IGF-1 treatment was associated with an increase in activated (Ki67^+^) Th2 cells (*p* < 0.0001; Figure 4i), primarily α_4_β_7_^+^CCR5^−^ (*p* = 0.020; Figure 4j). Accordingly, at the end of immunization (week 13), there was a drastic decrease in the frequency of α_4_β_7_^+^CCR5^+^ double-positive CD4^+^ cells in blood (*p* = 0.003; Figure 4k).

IGF-1 treatment also resulted in a lower frequency of mucosal vaccine-induced CD4^+^ T-cells (collected at week 9) expressing CCR5, as compared to macaques who received the vaccine only (*p* = 0.040; Figure 4l).

### 3.6. IGF-1 Treatment Does Not Affect DNA/ALVAC/gp120/alum Vaccine Efficacy

The IGF-1 treatment of DNA/ALVAC/gp120/alum-vaccinated animals augmented antibody responses to V1/V2, an envelope region associated with both protective and interfering antibodies, resulting in decreased V2-specific ADCC. We also observed an increase in protective responses such as CD14^+^ cell frequency and gut-homing CD4^+^ cells expressing low or no CCR5.

These data prompted us to investigate how IGF-1 treatment affected vaccine efficacy. Five weeks following the last immunization (week 17), all animals were exposed weekly to 11 repeated low doses of SIV_mac251_ challenges administered intravaginally (Figure 3a). Vaccination equally reduced the risk of SIV acquisition with or without IGF-1 compared to controls (*p* = 0.006 for vaccine+IGF-1, and *p* = 0.002 for vaccine; Figure 5a) with average vaccine efficacy at each exposure of 71% and 69%, respectively. Viral acquisition did not differ between the two vaccinated groups; however, the study was not powered to assess the difference between the two vaccinated groups. No significant difference in plasma virus levels was observed over time in the Vaccine and Vaccine+IGF-1 groups compared to controls (Figure 5b).

Interestingly, the vaccine group showed only a delayed CD4^+^ T-cell loss compared to control animals in week 2 (*p* = 0.012; Figure 5c). This group accordingly had a decreased copy number of SIV-DNA in the vaginal mucosa, but not when vaccination was combined with IGF-1 treatment (*p* = 0.045; Figure 5d). Given that the coadministration of DNA/ALVAC/gp120/alum vaccine with IGF-1 resulted in seemingly positive and negative effects on different immune responses, we investigated how IGF-1 impacted the immune correlates of the risk of virus acquisition. First, we focused on ADCC killing, one of the main correlates of the decreased risk of acquisition of the ALVAC-based vaccine strategies [6,30]. Although the Ab responses mediating ADCC did not differ quantitatively between the two groups, interestingly, the ADCC levels correlated with decreased risk of acquisition in DNA/ALVAC/gp120-vaccinated animals, but not in the animals treated with IGF-1 (*p* = 0.042, ρ = 0.58 and *p* = 0.884, ρ = −0.048; Figure 5e).

While the ADCC levels were similar in the two groups, the assay only tests the plasma, and the effector cells are not analogous, so any potential effect of IGF-1 on the effector cells would not be detected. We next tested the effect of the IGF-1 on ADCC effector cells through analyzing the ability of NK cells that express the NKG2A receptor to secrete granzyme B, perforin, IFN-γ, or TNF-α in vitro in the presence and absence of IGF-1. PBMCs collected from vaccinated macaques were left unstimulated or stimulated in vitro with a pool of gp120 peptides, or Phorbol 12-myristate 13-acetate (PMA) and Ionomycin (I; collectively, PMA+I), in the presence and absence of human recombinant IGF-1. The analysis revealed that, following stimulation with gp120 peptides, costimulation with IGF-1 decreased the frequency of NKG2A^+^ NK cells producing granzyme B, perforin, IFN-γ, and TNF-α (*p*-values = 0.008, 0.008, 0.016 and 0.008, respectively; Figure 5f). In contrast, when PBMCs were left untreated or stimulated with PMA+I, IGF costimulation did not affect the frequencies of NKG2A^+^ NK cells producing the different effector proteins (Appendix A). Interestingly, in the IGF-1-treated groups, the level of trogocytosis inversely correlated with ADCC (*p* = 0.010, ρ = −0.73 Figure 5g and Appendix A), though not in the untreated group.

In multiple macaque studies using ALVAC-based vaccines, the frequency of classical monocytes 1 week following vaccination was correlated with a reduced risk of SIV acquisition [6,29]. Similarly, in the vaccine group in this study, the frequencies of both classical and intermediate monocytes following vaccination correlated with decreased virus acquisition (*p* = 0.033, ρ = 0.60 in Figure 5h; *p* = 0.038, ρ = 0.59 in Figure 5i). In the vaccine+IGF-1 group, however, the correlations of these monocyte subsets were not significant (*p* = 0.775, ρ = 0.094 in Figure 5h; *p* = 0.592, ρ = −0.17 in Figure 5i).

In a prior study, we demonstrated the ability of the DNA/ALVAC/gp120 vaccination strategy to shape the innate immune response of monocytes through altering chromatin accessibility in CD14^+^ cells and enhancing their capability to perform efferocytosis [6]. Efferocytosis, a process that eliminates apoptotic cells in order to maintain tissue homeostasis [44], has a critical role in mediating the efficacy of the DNA/ALVAC/gp120 vaccine strategy [6]. Our data suggesting that IGF-1 increased CD14^+^ monocyte frequency was at odds with the lack of a correlation of CD14^+^ cells with the risk of virus acquisition. Instead, we considered the possibility that there were changes to monocyte functionality. To test this hypothesis, we performed an efferocytosis assay in vitro in the presence or absence of IGF-1. CD14^+^ cells isolated from PBMCs collected from naïve macaques were cocultured with apoptotic neutrophils in media either containing or not containing IGF-1. At the end of incubation, both the frequency of CD14^+^ cells that engulfed the apoptotic neutrophils and the amount of apoptotic cell cargo, expressed as the mean fluorescence intensity (MFI) of CFSE-neutrophils present in the engulfing CD14^+^ cells, were assessed via flow cytometry. Although the presence of IGF-1 did not change the frequency of CD14^+^ efferocytes in vitro (Appendix A), the cargo of apoptotic neutrophils engulfed by CD14^+^ efferocytes was lower in the coculture performed in the presence of IGF-1 than in the unstimulated condition (*p* = 0.016; Figure 5j), suggesting the ability of IGF-1 to influence the efficiency of CD14^+^ cells in conducting efferocytosis.

## 4. Discussion

The RV144 HIV vaccine trial in Thailand of volunteers at low risk of HIV acquisition (0.4% in women) demonstrated that the ALVAC-based HIV vaccine modality, together with two gp120 protein boosts in alum, was able to decrease the risk of HIV acquisition by 31.2% [2]. The correlates of decreased risk were the levels of antibodies to the envelope variable region 2 (V2) of antibody-dependent cell cytotoxicity (ADCC) in individuals with low anti-envelope IgA [39] and antigen-specific CD4^+^ cells producing IL-4 and IL-13 [42]. However, despite early excitement over the results of RV144, the shortcomings of the strategy became pronounced more recently with the failure of the HVTN-702 trial in a high-risk (4%) population in South Africa [45]. This trial, concluded in 2021 due to its futility, tested the same vaccine regimen used in RV144 against HIV clade C immunogens, with the important distinction that the two gp120 protein boosts were formulated with the more inflammatory, and more immunogenic, MF59 adjuvant [45]. While differences in the population’s risk of HIV infection and genetics and the co-morbidities of the volunteers enrolled in the studies have been evoked to explain the different outcome of RV144 and HVTN-702, insufficient attention has been given to the use of two different adjuvants in the trials.

We have partly elucidated the role of each adjuvant in the non-human primate (NHP) model. Using an identical ALVAC-based SIV vaccine with gp120 boosts adjuvanted either in alum (mirroring RV144) or MF59 (HVTN-702), we demonstrated that this vaccine regimen decreases the risk of acquisition of the highly pathogenic SIV_mac251_ only when the gp120 protein boost was formulated in alum. In an otherwise identical protocol, immunization with the MF59 adjuvant failed to protect macaques from SIV infection, despite being more immunogenic overall [5]. The use of MF59 as an adjuvant leads to decreased plasmablasts homing to the mucosa in both humans and macaques, which we found resulted in the induction of a lower level of mucosal antibodies to V2 in macaques.

While considering the pivotal impact of adjuvant choice on the efficacy of the ALVAC-based modality we observed in macaques, we also cannot use the negative results of HVTN-702 to dismiss the platform’s potential for use in humans. Firstly, a parallel trial using alum, which would have controlled for population genetics and co-morbidities, was not performed in South Africa; secondly, the antibody response to V2 induced in the HVTN-702 volunteers was low, impairing further testing of this primary immune correlate of risk identified in the RV144 trial.

Over the last decade, our investigations using the NHP model have significantly improved our understanding of the correlates of risk of HIV/SIV infection and the corresponding mechanisms of protection of the ALVAC-based vaccine, and we have used these advances to prepare for the potential optimization of this platform for an effective human vaccine. We achieved a reproducible and durable decreased risk of virus infection in vaccinated macaques using the most genetically diverse pathogenic strain of SIV_mac251_ infection [5,30] and convincingly established the robustness and relevance of the animal model. Importantly, we uncovered novel and reproducible innate and adaptive immune responses in this model and demonstrated that protection from infection requires a balance of immune responses that have opposite effects on the risk of virus infection. The combination of these vaccine modalities first causes inflammasome activation in monocytes, but this is then dampened through the induction of tolerogenic dendritic (DC-10) cells, finally resulting in the recruitment of anti-inflammatory monocytes to mediate anti-inflammatory efferocytosis, clear apoptotic cells, and maintain tissue homeostasis. All of these responses reproducibly correlate with a decreased risk of virus acquisition. Likewise, the priming of B-cells with properly engineered virus-like particles able to present the helical conformation of V2 (optimized via the deletion of V1, which elicits interfering antibodies) [30] results in V2-specific ADCC that induces the apoptosis of virus-infected cells [30]. The combination of both ADCC and efferocytosis is likely essential to clear virus-infected cells without causing inflammation that would recruit de novo target cells. Other key elements for a decreased risk of virus infection are mucosal NKp44^+^ cells producing IL-17 [5] that maintain tissue homeostasis and vaccine-induced gut-homing CD4^+^ T-cells expressing low or no CCR5 [29].

In contrast, the level of interfering antibodies to V1, the mucosal IFN-γ^+^NKp44^−^NKG2A^−^ cells, and the gut-homing activated CD4^+^ T-cells expressing CCR5 are all associated with increased virus acquisition. Thus, a finely tuned balance of complex pro-inflammatory and anti-inflammatory immunity is at the basis of the efficacy of the DNA/ALVAC/gp120/alum vaccine approach. It is significant that the involvement of each of these cell types has been corroborated by transcriptome, RNA-seq, ATAC-seq, and plasma proteome analyses of tissues of vaccinated macaques and functional studies in vaccinated human volunteers [5,6,29,46].

In one of the early studies in NHPs, we identified a 12-gene signature predictive of risk of virus infection with most genes (10 genes) belonging to the RAS pathway. The expression levels of those genes correlated with immune responses associated with a decreased risk of virus acquisition, such as anti-V2 antibody responses and IL-17^+^NK-p44^+^ cells, or with an increased risk, such as IFN-γ^+^NKp44^−^NKG2A^−^ cells [5]. We therefore hypothesized that the use of a potent inducer of RAS, such as IGF-1, could affect these responses and possibly improve the efficacy of ALVAC-based vaccines.

We demonstrate here that administration of DNA expressing IGF-1 during the DNA priming phase, and administration of the IGF-1 recombinant protein, Increlex^®^, during all immunizations significantly modulated the immunity elicited by the SIV-based DNA/ALVAC/gp120/alum vaccine, but it did not increase its efficacy. In vivo IGF-1 treatment, combined with vaccination, altered the antibody responses to V1/V2 epitopes and increased the level of mucosal IgG antibodies to conformational V2. Antibodies targeting the V1a region were previously identified as being capable of both inhibiting anti-V2 antibody-mediated cytotoxicity and reversing the ability of anti-V2 antibodies to block V2-α_4_β_7_ interaction [30]. Indeed, in animals treated with IGF-1, the levels of V2-specific ADCC decreased. The discrepancy between the increased antibody response to V1/V2 epitopes and decreased V2-specific ADCC may be due to the difference in the sections of V1/V2 recognized by the two assays. Whereas the antibody assay measures the antibody targeting of both the V1 and V2 regions, the V2-ADCC focuses on V2 epitopes only. Additionally, the correlation between specific ADCC against SIV-infected cells and the risk of SIV acquisition was ablated. This effect was apparently due not to an increase in trogocytosis in IGF-1 treated animals but possibly to a direct effect of IGF-1 on NK function. In fact, IGF-1 promotes both the development and the cytotoxic activity of NK cells in humans [47]. Surprisingly, we show here that in vaccinated macaques, NKG2A^+^ NK cells stimulated with gp120 peptides in the presence of IGF-1 decrease their cytocidal functions through decreasing the production of Granzyme B, Perforin, IFN-γ, and TNF-α.

Strikingly, IGF-1 treatment in vivo significantly increased systemic CD14^+^CD16^−^ classical monocytes at the expense of intermediate and non-classical monocytes and decreased CD14^+^ cell trogocytosis and efferocytosis in vitro. Although classical monocytes were consistently correlated with a reduced risk of SIV acquisition in prior studies [6,29], the impact of IGF-1 on their frequency may not be sufficient to ameliorate the vaccine efficacy when not associated with the functional reprogramming of monocytes. Indeed, in-depth analyses of CD14^+^ cells conducted in a prior study of the DNA/ALVAC/gp120/alum vaccine based on immunogens deleted in the V1 region of gp120 demonstrated that the regimen was able to modify the expression of genes involved in CREB pathway activation and increase the chromatin accessibility of a site in the enhancer region of the CREB-1 gene [6], suggesting a need for the reprogramming of monocytes towards an anti-inflammatory status to support vaccine efficacy. The induction of protective classical monocytes contrasted with the lack of an increase in vaccine efficacy and may be due to IGF-1 affecting chromatin remodeling. Multiple studies have shown that the activation of the IGF-1 receptor pathway, which leads to the activation of mTOR and glycolysis, can induce trained immunity and trigger the expression of pro- and anti-inflammatory genes [48,49,50]. Therefore, IGF-1 may compromise the vaccine-induced modification of chromatin accessibility necessary to mediate effective efferocytosis to properly eliminate apoptotic cells and maintain tissue homeostasis [44,51]. Efferocytosis has previously been identified as a correlate of vaccine efficacy [6], and in the current study, we demonstrate that IGF-1 diminishes the ability of CD14^+^ cells to engulf apoptotic cells. Indeed, IGF-1 secreted by macrophages can also decrease the efferocytic capacity of larger apoptotic cells executed by non-professional phagocytes, such as epithelial cells; however, this effect was not identified directly in macrophages [52,53].

Our study corroborates the idea that an effective anti-HIV vaccine must elicit balanced pro- and anti- inflammatory responses along both qualitative and quantitative dimensions. Indeed, because the alum adjuvant engages monocytes rather than neutrophils as MF59 does, it contributes to maintaining a regulated balance between pro- and anti-inflammatory responses that are crucial for vaccine efficacy [6]. Similarly, IGF-1 can induce immune suppressive activity in monocytes [28] and causes changes that may be critical for regulating pro- vs. anti-inflammatory responses during neuroinflammation [54].

## 5. Conclusions

We demonstrate here that the coadministration of IGF-1 and SIV-based DNA/ALVAC/gp120/alum, on one hand, may have ameliorated vaccine efficacy through decreasing the gut-homing CD4^+^ Th2 cells expressing CCR5 yet, on the other hand, worsened other protective responses through diminishing V2-specific ADCC and the capability of CD14^+^ cells to perform efferocytosis (Figure 6). The balance of these opposite effects is likely responsible for the similar efficacy of the vaccine alone and the vaccine supplemented with IGF-1.

## Figures and Tables

**Figure 1 vaccines-11-01662-f001:**
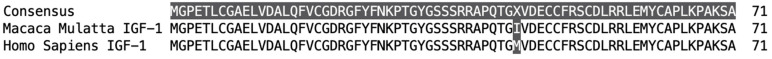
Geneious Prime^®^ alignment of the amino acid sequences of the IGF-1 proteins of rhesus macaque (*Macaca mulatta*) and human (*Homo sapiens*). The top row indicates consensus between the two sequences. The difference in position 44 (I—Isoleucine vs. M—Methionine) is highlighted.

**Figure 2 vaccines-11-01662-f002:**
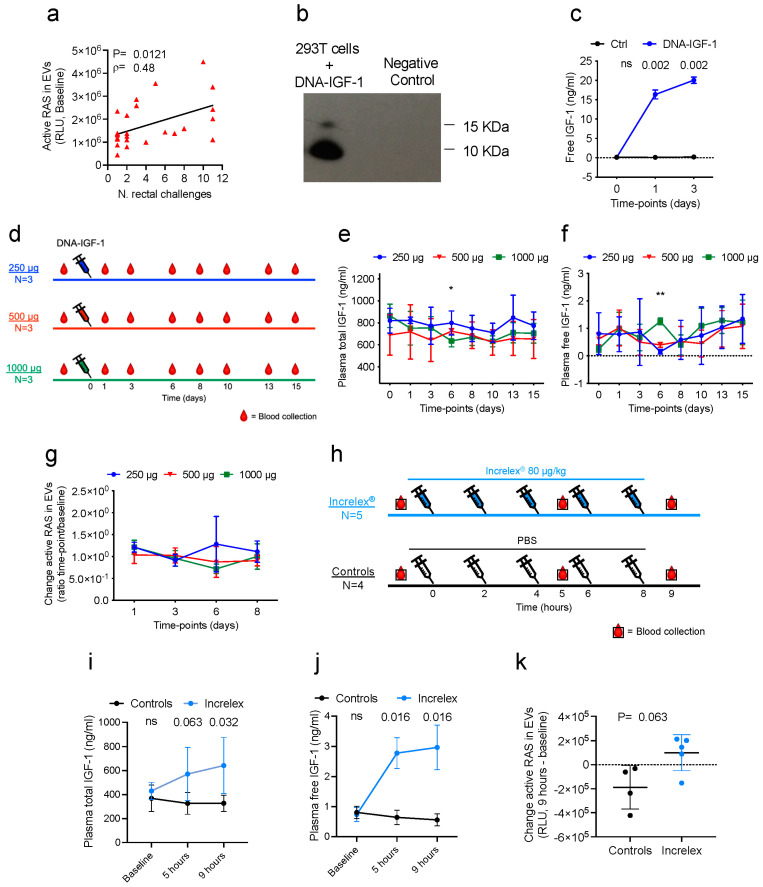
IGF-1 administration and RAS activation. (**a**) Two-tailed Spearman correlation and simple linear regression between the levels of active RAS (relative light units; RLU) in plasma extracellular vesicles (EVs) at baseline and the time of acquisition (TOA) in ALVAC-SIV/gp120/alum-vaccinated animals (*n =* 26) [5]. (**b**) SDS-PAGE analysis of the IGF-1 protein in the supernatant of 293T cells transfected with DNA-IGF-1 (left; densitometry band of interest: 59.8) or a negative plasmid (right; densitometry band of interest: −1.1). The molecular weight of 15 and 10 KDa ladder bands are indicated. (**c**) Free IGF-1 levels (ng/mL) in the supernatant of 293T cells transfected with DNA-IGF-1 (*n* = 6; blue) or a negative plasmid (*n* = 6; black) at baseline and 1 and 3 days following transfection. (**d**) Schematic study design of plasmid DNA-IGF-1 administration and blood sample collections in three groups of 3 animals each (250 μg [blue], 500 μg [red], and 1000 μg [green]). (**e**,**f**) Plasma levels of (**e**) total and (**f**) free IGF-1 (ng/mL) at baseline and following DNA-IGF-1 administration in macaques administered different amounts of plasmid. (**g**) Variation in levels of active RAS in plasma EVs isolated from macaques administered different amounts of DNA-IGF-1 plasmid. For each animal, the variation at each timepoint was calculated as the ratio with the baseline level. (**h**) Schematic study design of 5 Increlex^®^ (cerulean; *n* = 5) or PBS (black; *n* = 4) administrations and blood sample collections in macaques. Blood was collected at baseline and following 3 and 5 administrations. (**i**,**j**) Plasma levels of (**i**) total and (**j**) free IGF-1 (ng/mL) at baseline and following 3 and 5 Increlex^®^ or PBS administrations. (**k**) Variation in levels of active RAS (RLU) in plasma EVs isolated from macaques administered of 5 Increlex^®^ or PBS injections. For each animal, the variation was calculated as the subtraction of the active RAS levels following 5 injections and at baseline. Comparisons: (**c**,**i**–**k**) the comparisons between the groups were performed as two-tailed Mann–Whitney U tests at each timepoint; (**e**–**g**) the comparisons between the three groups were performed as Kruskal–Wallis ANOVA tests at each timepoint. The mean and SD are displayed. Statistical significance: * *p* < 0.05; ** *p* < 0.01. (**c**,**f**,**k**) dotted lines represent the zero value. (**d**,**h**) red triangular drops represent blood collections.

**Figure 3 vaccines-11-01662-f003:**
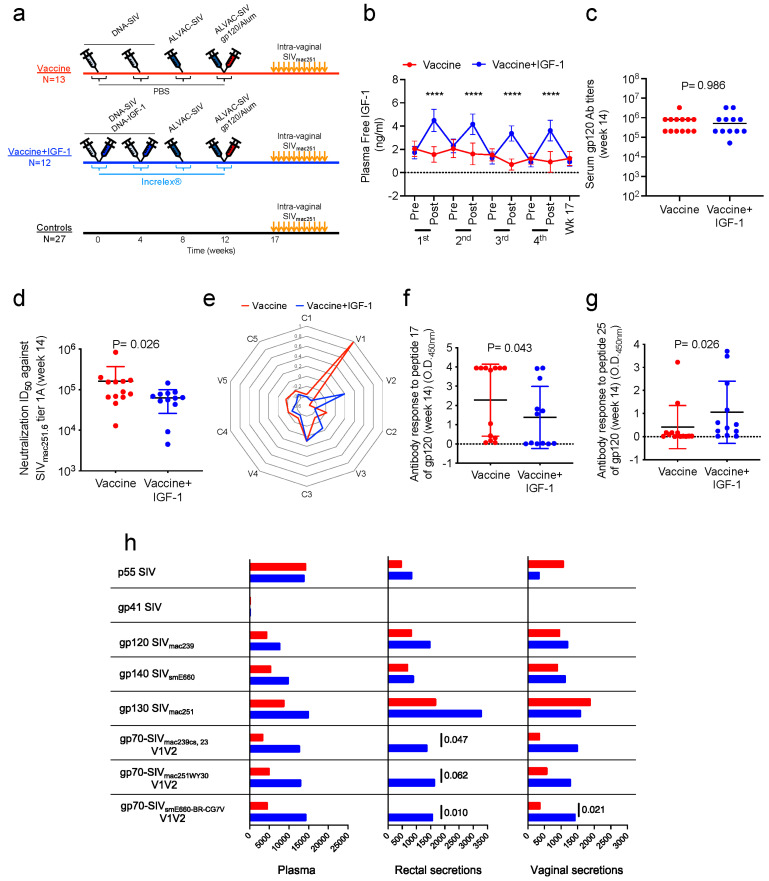
Vaccination in association with IGF-1 administration and antibody responses. (**a**) Schematic study design with immunization schedule, in association (blue) or not (red) with DNA-IGF-1 and Increlex^®^ administration (weeks 0–12) and SIV_mac251_ challenges (weeks 17–27). (**b**) Plasma levels of free IGF-1 (ng/mL) in animals vaccinated in association or not with IGF-1 administration. Levels were tested prior to and following each immunization, and at the first challenge (week 17). (**c**) Serum IgG antibody titers to whole SIV_m766_ gp120 in vaccinated (*n* = 13) and vaccinated+IGF-1 (*n* = 12) animals at week 14. (**d**) Neutralizing antibody responses (ID_50_) to tier-1A SIV_mac251_._6_ in serum of vaccinated (*n* = 13) and vaccinated+IGF-1 (*n* = 12) animals in week 14. (**e**) Radar plots showing the antibody responses to constant and variable regions of SIV_m766_ gp120 in serum of vaccinated (*n* = 13) and vaccinated+IGF-1 (*n* = 12) animals in week 14. Data are plotted as median of normalized z-scores for each region across all the animals. Comparisons were performed using the Mann–Whitney U test on the absolute values (optical densities, ODs) between vaccine and vaccine+IGF-1 groups. (**f**,**g**) Antibody response (OD) to peptides (**f**) 17 and (**g**) 25 of SIV_m766_ gp120 in serum of vaccinated and vaccinated+IGF-1 animals in week 14. (**h**) Summary graph representing the IgG antibody responses to p55 gag, gp41, and gp120 proteins and gp70 V1/V2 scaffolds in plasma, and rectal and vaginal secretions of vaccinated and vaccinated+IGF-1 animals in week 14. For vaginal secretions, 3 vaccinated animals and 4 vaccinated+IGF-1 animals were excluded due to blood contaminations. Data are represented as the median of the absolute values of the MFI for plasma and the specific activity for rectal and vaginal secretions. Comparisons were performed using the two-tailed Mann–Whitney U test on the absolute values between vaccine and vaccine+IGF-1 groups. Comparisons: (**b**,**h**) two-tailed Mann–Whitney U test with mean and SD; (**c**) two-tailed Mann–Whitney U test and median; (**d**,**f**,**g**) two-tailed Mann–Whitney U test and mean with SD. Statistical significance: **** *p* < 0.0001. (**b**,**f**,**g**) dotted lines represent the zero value.

**Figure 4 vaccines-11-01662-f004:**
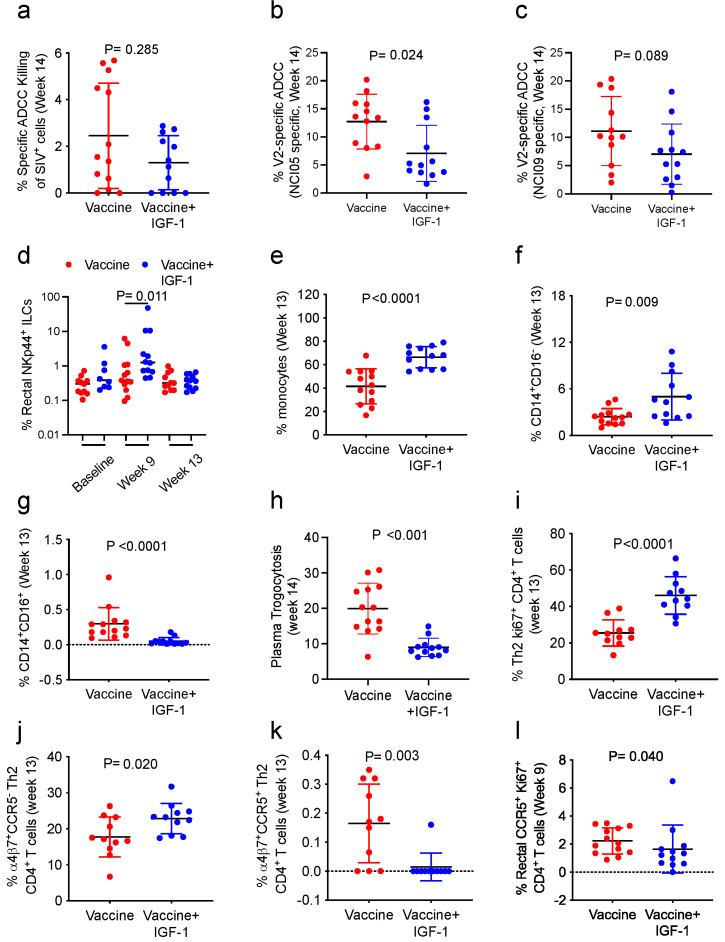
Effect of IGF-1 on ADCC, monocytes, and CD4^+^ T-cells. (**a**) Specific ADCC killing of SIV-infected cells in plasma of vaccinated (*n* = 13) and vaccinated+IGF-1 (*n* = 12) animals in week 14. (**b**,**c**) V2-specific ADCC against gp120-coated cells in plasma of vaccinated (*n* = 13) and vaccinated+IGF-1 (*n* = 12) animals in week 14. V2-specific ADCC was assessed using F(ab′)2 of (**b**) NCI05 and (**c**) NCI09 antibodies targeting V2. (**d**) Frequencies of innate lymphoid cells expressing NKp44 receptor in rectal mucosa of vaccinated and vaccinated+IGF-1 animals at baseline (*n* = 11 and *n* = 8, respectively), week 9 (*n* = 13 and *n* = 12), and week 13 (*n* = 11 and *n* = 12). (**e**–**g**) Frequencies of (**e**) total, (**f**) classical, and (**g**) intermediate monocytes in blood of vaccinated (*n* = 13) and vaccinated+IGF-1 (*n* = 12) animals in week 13. (**h**) Trogocytosis score in plasma of vaccinated (*n* = 13) and vaccinated+IGF-1 (*n* = 12) animals in week 14. (**i**–**k**) Frequencies of type 2 T helper CD4^+^ T-cells (CXCR3^−^CCR6^−^) expressing (**i**) Ki67, (**j**) α4β7 but not CCR5, and (**k**) α4β7 and CCR5, in blood of vaccinated (*n* = 11) and vaccinated+IGF-1 (*n* = 11) animals following last immunization (week 13). (**l**) Frequency of Ki67^+^ CD4^+^ T-cells expressing CCR5 assessed in rectal mucosa of vaccinated (*n* = 13) and vaccinated+IGF-1 (*n* = 12) animals following the 3rd immunization (ALVAC-SIV boost; week 9). Comparisons: (**a**–**c**,**e**–**l**) two-tailed Mann–Whitney U test with mean and SD; (**d**) two-tailed Mann–Whitney U test and median (unadjusted *p* values). (**g**,**k**,**l**) dotted lines represent the zero value.

**Figure 5 vaccines-11-01662-f005:**
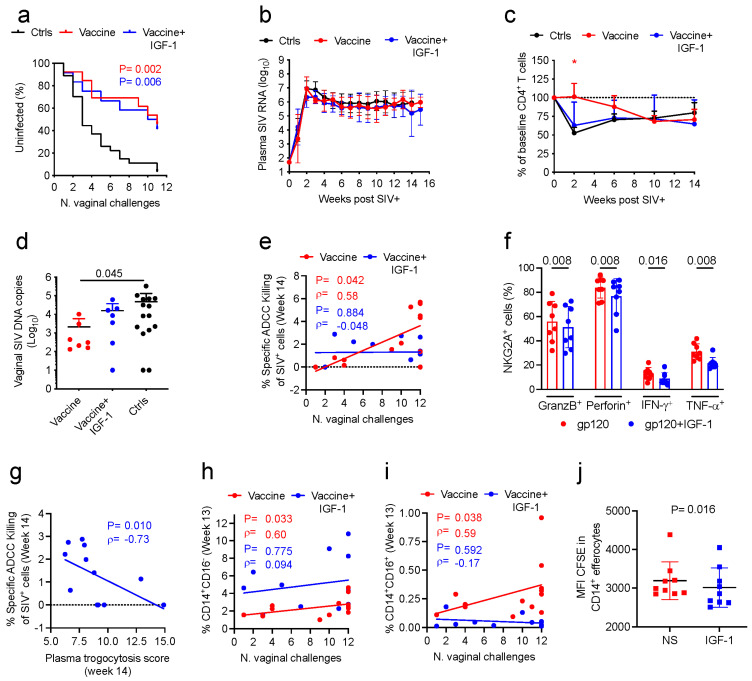
Effect of IGF-1 on vaccine efficacy and correlates of risk of viral acquisition. (**a**) SIV_mac251_ acquisition. The number of intravaginal exposures before infection was assessed in animals vaccinated with IGF-1 (*n* = 13) or without IGF-1 (*n* = 12) and relative to controls (*n* = 27; Log-rank Mantel-Cox test). (**b**) SIV RNA levels in plasma over time following SIV_mac251_ infection (weeks; geometric mean with error and 95% CI) in vaccinated (*n* = 7), vaccinated+IGF-1 (*n* = 7), and control (*n* = 26) animals. (**c**) Percentage of CD4^+^ T-cell changes in blood over time following SIV_mac251_ infection (weeks; mean ± s.e.m.) in vaccinated (*n* = 7), vaccinated+IGF-1 (*n* = 7), and control animals (*n* = 18). Asterisk (*) indicates two-tailed Mann–Whitney comparison test between the vaccine (red *) or vaccine+IGF-1 (none) groups and the control group *p* < 0.05 or *p* > 0.05, respectively. (**d**) Log_10_ of SIV-DNA copies in vaginal mucosa 2–3 weeks after infection in vaccinated (*n* = 7), vaccinated+IGF-1 (*n* = 7), and control (*n* = 15) animals. (**e**) Correlations between the specific ADCC killing of SIV-infected cells in plasma of vaccinated (*n* = 13; red) and vaccinated+IGF-1 (*n* = 12; blue) animals in week 14 and at the time of acquisition (TOA). (**f**) Frequencies of NKG2A^+^ NK cells expressing Granzyme B, Perforin, IFN-γ, or TNF-α following gp120 or gp120+IGF-1 stimulation in PBMCs collected from vaccinated animals (*n* = 8) following last immunization (week 14). Comparisons: two-tailed Wilcoxon signed rank test between gp120 and gp120+IGF-1 treated cells for each secreted protein (unadjusted *p* values) with mean and SD. (**g**) Correlation between the specific ADCC killing of SIV-infected cells in plasma of vaccinated+IGF-1 (*n* = 12) animals in week 14 and the Trogocytosis score in plasma in week 14. (**h**,**i**) Correlations between the frequencies of (**h**) classical or (**i**) intermediate monocytes in blood of vaccinated (*n* = 13; red) and vaccinated+IGF-1 (*n* = 12; blue) animals in week 13 and the TOA. (**j**) Levels of apoptotic neutrophils engulfed by CD14^+^ efferocytes following 24 h in vitro incubation without (NS) or with IGF-1 stimulation (IGF-1) in CD14^+^ cells isolated from naïve macaques (*n* = 9). The levels are expressed as the MFI of CFSE used to mark the neutrophils. Comparisons: (**d**) two-tailed Mann–Whitney U test and mean with SD; (**f**,**j**) two-tailed Wilcoxon signed rank test with mean and SD. Correlations: (**e**,**g**–**i**) two-tailed Spearman correlation with simple linear regression. Statistical significance: * *p* < 0.05. (**c**,**e**,**g**) dotted lines represent the zero value.

**Figure 6 vaccines-11-01662-f006:**
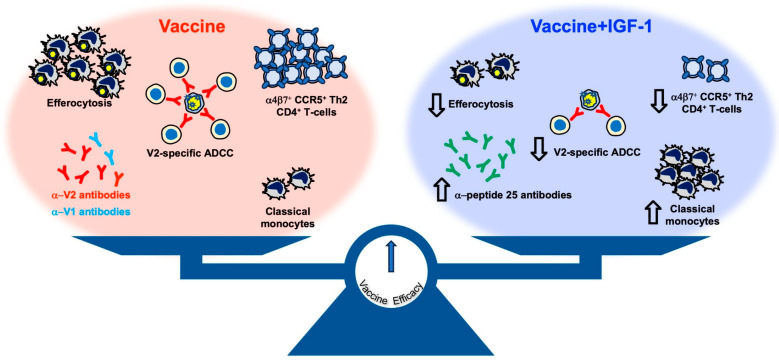
Schematic representation of immune responses increased (up arrow) and decreased (down arrow) through the coadministration of IGF-1 and anti-SIV DNA/ALVAC/gp120/alum vaccine. The left side of the image portrays correlates of decreased/increased risk of SIV acquisition identified in prior and current DNA/ALVAC/gp120/alum vaccine studies, whereas the right side represents the effect of the coadministration of IGF-1 and the vaccine on the same correlates. The balance between the increase and decrease in immune correlates due to IGF-1 administration results in a similar vaccine efficacy in vaccinated and vaccinated+IGF-1 non-human primates.

## Data Availability

The original data and files used to generate the data presented in this study, as well as a table reporting data used to generate each figure of this manuscript (organized by figure), are available on the Zenodo repository (https://zenodo.org/; https://zenodo.org/records/8387855).

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
