# Peer review of "In Vivo Treatment with Insulin-like Growth Factor 1 Reduces CCR5 Expression on Vaccine-Induced Activated CD4+ T-Cells"

_vaccines, 2023, doi:10.3390/vaccines11111662_

Round 1

Reviewer 1 Report

Comments and Suggestions for Authors

This paper deals with the analysis of some immunological parameters which are relevant for the characterization of the immune response to an immunization protocol to SIV infection in a macaque experimental model for human HIV.

            The paper is excellently written and presented. It includes some pertinent and well designed figures. Materials and Methods are extensively and comprehensively detailed, describing a good number of different analytical procedures.

            The authors are experts in the field and have already made previous sound contributions.

            The final conclusion of the paper is in agreement with the analysis of the reported observations and in line with the discussion posed by the authors.

Author Response

We are grateful to the reviewer for the kind comments and for reviewing the manuscript.

Reviewer 2 Report

Comments and Suggestions for Authors

This manuscript by Bissa et al. presents the interesting observation that IGF-1 both modulates the nature of anti-HIV vaccine responses, and unexpectedly, downmodulates CCR5 on gut-homing CD4 T cells, raising the possibility that IGF-1 could help restrain the HIV CD4 T cell reservoir. The experiments presented involving IGF-1 treatment and immunization of monkeys, are impressive in their scope and ambitions and represent an enormous amount of work. Given previous work on Ras signaling and neutralizing antibody responses, the detailed observations of the complex effects of IGF-1 treatment on different aspects of anti-SIV immunity in vivo are important results for the HIV vaccine community. The lack of effect of IGF-1 on vaccine-induced protection against infection are negative results that are nonetheless important for the field in order to reduce the need for other related experiments thus preventing unnecessary, time consuming and costly monkey experiments. Furthermore, the observation of the effects of IGF-1 in modulation CCR5 positivity in a subset of lymphoid cells has important potential clinical significance worthy of reporting.

Overall, this is well-written study describing extensive in vivo experiments on modulation of SIV/HIV vaccine efficacy through altering Ras pathways by IGF-1 and demonstrating the effects of IGF-1 on CCR5+ lymphocytes. 

One point of interest is that the reduction of CCR5+ CD4 cells by the addition of IGF-1 to the vaccine (Fig. 3k, l) is suggested to have the potential reduce the SIV/HIV reservoir. The observation that (Fig. 4d) vaccine + IGF-1 may have higher vaginal SIV DNA copies than the vaccine alone might be interpreted to contradict this predictiont. How do the authors understand these results in terms of the possibility that IGF-1 may reduce the viral reservoir? Is there any data from the viral challenge in Figure 4 on rectal SIV DNA copies, to correspond to the rectal CCR5- CD4 cells seen in Figure 3, and/or conversely on levels of vaginal CCR5- CD4 cells from the experiment of Figure 3?

Minor Comments:

1.  Figure 1 e, f showing the increase at Day 6 at the highest dose of DNA-IGF-1 is interesting. It would be helpful to know if the authors are aware of any literature on the day to day fluctuation of IGF-1 in plasma under normal conditions in macaques or humans, since there are no untreated control monkeys. In other words, by way of conclusion, was the DNA inoculation successful at all in altering blood levels of IGF-1 (i.e. at the highest dose). By contrast, the Increlex data have the needed controls for comparison and has sound data for IGF1 accumulation and strongly suggestive data in support of effects on Ras signaling.

2. It is worth building a quick review of the “tier” nomenclature of virus neutralization into the introduction of the neutralization data in Figure 2.

3. Much of the data point to important modulations of monocyte functions. Could the authors comment on whether another monocyte functions were examined for functional alterations with the combined vaccine – IGF-1 treatment and if so, what were the results?

Author Response

We are grateful to the reviewer for the kind comments and for reviewing the manuscript. We respond to each comment individually as follows. 

One point of interest is that the reduction of CCR5+ CD4 cells by the addition of IGF-1 to the vaccine (Fig. 3k, l) is suggested to have the potential reduce the SIV/HIV reservoir. The observation that (Fig. 4d) vaccine + IGF-1 may have higher vaginal SIV DNA copies than the vaccine alone might be interpreted to contradict this prediction. How do the authors understand these results in terms of the possibility that IGF-1 may reduce the viral reservoir? Is there any data from the viral challenge in Figure 4 on rectal SIV DNA copies, to correspond to the rectal CCR5- CD4 cells seen in Figure 3, and/or conversely on levels of vaginal CCR5- CD4 cells from the experiment of Figure 3?

We agree with the reviewer that the administration of the IGF-1 in association with vaccination is not able to decrease the viral load. In fact, the viral load measured in plasma of vaccinated and vaccinated+IGF-1 animals that acquired SIV following the challenge phase did not differ. Additionally, the DNA proviral load measured in the mucosa during the acute phase of infection is not significantly different between animals that received the vaccine with or without the IGF-1.

We agree with the reviewer that our speculation should be supported by more evidence and therefore we have decided to remove it from the manuscript.

Minor Comments:

1.Figure 1 e, f showing the increase at Day 6 at the highest dose of DNA-IGF-1 is interesting. It would be helpful to know if the authors are aware of any literature on the day to day fluctuation of IGF-1 in plasma under normal conditions in macaques or humans, since there are no untreated control monkeys. In other words, by way of conclusion, was the DNA inoculation successful at all in altering blood levels of IGF-1 (i.e. at the highest dose). By contrast, the Increlex data have the needed controls for comparison and has sound data for IGF1 accumulation and strongly suggestive data in support of effects on Ras signaling.

In response to the reviewer’s comments, we repeated an extensive research in the literature. We were not able to find any data reporting the fluctuation of IGF-1 in macaques in a period of multiple days similar to what is reported in the manuscript. Unfortunately, generally the research both in humans and in macaques focuses either on the circadian fluctuation of the IGF-1 or its change across the different phases of life. However, we found a study by Juul et al. (https://pubmed.ncbi.nlm.nih.gov/9402266/) that was conducted in humans and reported the monthly fluctuations of IGF-1 based on the menstrual cycle. We have cited this work in the manuscript and clarified that the results of the DNA-IGF-1 administration may be affected by confounding factors (L602).

In text, “However, these differences may be due to normal fluctuation of IGF-1 levels, as reported in humans (Juul 1997), rather than as a consequence of the administration of different amounts of DNA-IGF-1.”

  1. It is worth building a quick review of the “tier” nomenclature of virus neutralization into the introduction of the neutralization data in Figure 2.

 We thank the reviewer for the suggestion. We have clarified the “tier” nomenclature by adding the following sentence (L665-7).

In text, “Next, we measured the serum levels of neutralizing antibodies recognizing SIV viruses with different sensitivity to antibody-mediated neutralizations (Tiers).”

  1. Much of the data point to important modulations of monocyte functions. Could the authors comment on whether another monocyte functions were examined for functional alterations with the combined vaccine – IGF-1 treatment and if so, what were the results?

We thank the reviewer for the constructive comment. In the manuscript we have demonstrated that the IGF-1 is able to modify the efferocytic activity of CD14+ monocytes (Fig. 4j), as well as affect antibody-mediated trogocytosis (Fig. 3h). Unfortunately, the lack of an adequate number of cryopreserved PBMCs isolated from vaccinated and vaccinated+IGF-1 animals enrolled in the macaque study did not allow us to conduct further functional assays on monocytes. We deeply appreciate the reviewer’s suggestion, and we will consider it for future studies, in case we have the opportunity to further pursue the use of IGF-1 in vaccine studies conducted in macaques. We hope that the reviewer will be satisfied with what was reported in the manuscript.

Round 2

Reviewer 2 Report

Comments and Suggestions for Authors

The authors have provided substantive and adequate responses to my previous comments.